# Cutaneous and Mucosal Melanomas of Uncommon Sites: Where Do We Stand Now?

**DOI:** 10.3390/jcm10030478

**Published:** 2021-01-28

**Authors:** Emi Dika, Martina Lambertini, Cristina Pellegrini, Giulia Veronesi, Barbara Melotti, Mattia Riefolo, Francesca Sperandi, Annalisa Patrizi, Costantino Ricci, Martina Mussi, Maria Concetta Fargnoli

**Affiliations:** 1Dermatology, IRCCS Policlinico di Sant’Orsola, via Massarenti 9, 40138 Bologna, Italy; mlambertini@hotmail.it (M.L.); giulia.veronesi.md@gmail.com (G.V.); annalisa.patrizi@unibo.it (A.P.); mussi.martina1809@gmail.com (M.M.); 2Dermatology, Department of Experimental, Diagnostic and Specialty Medicine (DIMES), University of Bologna, 40138 Bologna, Italy; 3Dermatology, Department of Biotechnological and Applied Clinical Science, University of L’Aquila, 67100 L’Aquila, Italy; cristina.pellegrini@univaq.it (C.P.); mariaconcetta.fargnoli@univaq.it (M.C.F.); 4Division of Oncology, IRCCS di Policlinico Sant’Orsola, via Massarenti 9, 40138 Bologna, Italy; barbara.melotti@aosp.bo.it (B.M.); francesca.sperandi@aosp.bo.it (F.S.); 5Department of Experimental, Diagnostic and Specialty Medicine, University of Bologna, 40138 Bologna, Italy; mattia.riefolo@unibo.it (M.R.); costanricci@gmail.com (C.R.); 6Pathology Unit, Ospedale Maggiore, 40100 Bologna, Italy

**Keywords:** melanoma, nail, oral, scalp, genital, acral

## Abstract

Melanomas arising at uncommon sites include a group of lesions related to unusual localizations in specific ethnic groups. The rarity of the disease often represents a limit to the participation of patients in specific trials. However, this peculiar genetic scenario has important therapeutic implications regarding new oncologic therapies. The aim of this article is to review the clinical features, somatic alterations and therapeutic options for melanomas of uncommon sites. They can be classified as cutaneous and mucosal lesions affecting the nail apparatus, palms/soles, oral mucosa, genital area and scalp. The prognosis may be worse compared to melanomas of other districts, and a prompt diagnosis may dramatically influence the outcome. Dermatologists and oncologists should therefore distinguish this melanoma subgroup in terms of surgical intervention and medical treatment. Due to the lack of mutations in genes usually found in cutaneous melanomas, the discovery of novel targets is required to develop new strategies and to change the prognosis of non-responders or wild-type patients.

## 1. Introduction

Melanomas of uncommon sites encompass both cutaneous and mucosal lesions related to an unusual localization in specific ethnic groups. The histopathological interpretation may be challenging, and known site-related atypical features may sometimes increase diagnostic dilemmas for pathologists. They represent a separate subgroup with peculiar genetic alterations and, in some cases, may be associated with a worse prognosis compared to melanomas of other districts. The umbrella term “sun-protected melanomas” has been considered inadequate to describe a heterogeneous pool of mucosal, acral and vulvovaginal melanomas with distinct genetic profiles [1]. The treatment choice and outcome may therefore be challenging.

The aim of this article is to describe the clinical features of cutaneous and mucosal melanomas of uncommon sites and review the reported somatic molecular alterations and emerging treatment options for these aggressive variants. Dermatologists and oncologists should specifically consider this subgroup in terms of surgical intervention and medical treatment. Here, we will focus on melanomas arising on acral body areas, including the nail apparatus and palms/soles, oral mucosa, genital area and scalp. 

## 2. Cutaneous Melanomas at Uncommon Sites

### 2.1. Acral Melanomas 

The term acral melanoma includes a subset of melanomas affecting the nail unit, palms and soles. The pathogenetic mechanisms of acral melanoma have not yet been completely clarified [2]. A role for the trauma has been proposed, although, according to some authors, the trauma may simply represent the reason inducing patients to seek a prompt medical evaluation, with the consequent unexpected recognition of the disease [2,3,4]. The role of UV exposure is not fully applicable to acral melanomas, but it seems more relevant in cutaneous, as compared to nail unit, lesions. UV signature mutations (tandem CC>TT transitions and C>T transitions at dipyrimidine sites) are not commonly detected in acral melanoma [1,5,6]. However, in 1987, Baran and Juhlin reported that the nail plate may represent a convex lens favoring UV exposure to nail matrix melanocytes [7]. In addition, solar elastosis has been detected in nail melanoma samples, although this feature is not routinely reported by pathologists.

Nail melanoma is uncommon in White patients, accounting for only 2% of all melanomas but for almost one-fifth of the total cases in non-White subjects, including Asians and Afro-Americans [8,9]. It usually presents as a brownish/black/grayish irregular longitudinal band of pigmentation (nail melanonychia) that may be associated with nail plate dystrophy and abnormalities (Figure 1). The pigmentation can extend to the nail folds or to the surrounding skin with the so-called Hutchinson’s sign [10,11,12] (Figure 1 and Figure 2). Amelanotic lesions clinically present as a band of erythronychia or as eroded nodules with nail plate abnormalities showing polymorphous vessels and milky-red areas on dermatoscopy [13,14]. Nail melanoma mainly arises in the great toenail, the thumb and the second digit. Possible differential diagnoses include tinea unguium, warts, hemorrhages or pyogenic granulomas, often determining a delay in the diagnosis and, thus, a worse prognosis [15]. 

Cutaneous acral melanoma clinically appears as a pigmented brown-to-black macule (Figure 3) or nodule, sometimes ulcerated, although amelanotic lesions can also be observed. A parallel ridge pattern is a dermatoscopic feature suggestive for acral melanoma, together with polymorphous vessels [15,16]. 

Histopathological subtypes of acral melanoma most commonly include acral lentiginous melanoma, followed by nodular (20–39%) and superficial spreading types (8–27%). However, in a series of 31 Caucasian patients affected by acral melanoma, our group detected the nodular subtype in 48.4% of cases, and the acral lentiginous and superficial spreading subtypes in 22.6% each [17]. 

Surgical procedures and the ability to achieve wide margins can be difficult in acral melanomas due to the attempt to preserve tissue and maintain functional integrity. Incisions in digital and interdigital spaces must be carefully planned. In the literature, Mohs micrographic surgery is indicated as a possible treatment option for in situ melanoma of the nail [18]. Other authors described 14 patients with invasive nail unit melanoma treated with Mohs micrographic surgery with good results (6/14) but, also, with local recurrences (3/14), positive lymph nodes (1/14) and distal metastases (2/14) [18,19]. Amputation of the involved phalanx or digit is traditionally indicated, according to melanoma thickness, possibly with sentinel lymph node biopsy [20,21]. However, with regard to melanoma of the nail, some authors reported that disarticulation is not associated with a better prognosis [21]. A lymph node biopsy is an important prognostic element for acral melanoma, but there is no correlation between Breslow thickness and lymph node status [20]. 

Understanding the relationships of each subtype of acral melanoma with constitutional and genetic factors can be the key to developing advanced prevention strategies. Overall, acral melanomas show a markedly different genomic landscape from non-acral melanomas, with a lower mutation burden and large-scale structural rearrangements (deletions, duplications and tandem duplications) and genomic aberrations. Indeed, as reported by Hayward et al. (2017), single-nucleotide variants were about 18 times less frequent in acral (mean 2.64 mutations per megabase) than in cutaneous non-acral melanomas (mean of 49.17 mutations), while somatic structural variants (SVs) were approximately three times more prevalent (mean of 342.40 vs. 101, respectively) [22].

Overall, *BRAF*, *NRAS*, *KIT*, *MAP2K2* and *NF1* have been identified as significant driver genes in acral lesions; however, the mutation frequencies varied considerably compared to conventional cutaneous melanoma. Acral melanoma mainly exhibited a variety of “triple wild-type” mechanisms, including point mutations in *KIT* detected in up to 23% of cases and focal amplifications of *KIT*, *CCND1*, *CDK4*, *MDM2*, *KRAS* and *PDGFRA*. Instead, *BRAF* and *NRAS* mutations have been identified in up to 33% and 46%, respectively [17]. Recently, in a group of 31 White patients, we found lower mutation rates for *BRAF* and *KIT* of 12.9% and 20.0%, respectively, and a prevalence of *NRAS* mutations of 27.3% [19]. Interestingly, *BRAF* mutations were less likely to occur on volar or ungual melanomas than on dorsal acral skin, probably reflecting the increased role of UV exposure [23]. Finally, in further contrast to cutaneous melanomas, acral lesions infrequently carried *PTEN*, *DDX3X*, *RASA2*, *PPP6C*, *RAC1* or *RB1* pathway mutations [24,25]. 

SVs and copy number variations (CNV) were identified mainly in the *NF1*, *TP53*, *PTEN* and *KIT* genes, while structural rearrangements producing kinase fusions that may activate the MAPK pathway were reported in the *RAF1*, *RAC1*, *MAP2K2*, *MAP3K9*, *TRRAP* and *PAK1* genes [24]. *BRAF* fusion genes such as AGK-BRAF, ERC1-BRAF and CNTNAP2-BRAF that retained the BRAF kinase domain with a loss of the autoinhibitory domain were also identified [26]. Other fusion genes involved receptor tyrosine kinases, such as NTRK3 and ALK. Interestingly, the authors also observed 23% of acral melanomas with structural rearrangements or deletions affecting *NF1* [26].

The differences between the mutational statuses of cutaneous acral and nail melanomas have been reported in the COSMIC database, describing mutations of *BRAF, NRAS* and *KIT* in 14.6%, 8.8% and 11.9% of cutaneous acral melanoma cases, compared to 10.2%, 33.3% and 19% in nail melanomas, respectively [27]. Mutations in other genes have a significantly lower frequency, with acral lesions showing changes of *TP53* in 8.7% and *SKT11* in 4.3% of cases and nail melanomas harboring mutations of *GNA11* in 2.2%. The whole exome sequencing study performed by Lee et al. (2018) revealed additional differences, with cutaneous acral melanoma carrying *BRAF* and *NRAS* mutations while nail lesions showed a higher number of copy number alterations (median of 17 for acral vs. 42 for the nail apparatus). A genomic aberration of *CARD11*; chromatin remodeling genes (*ARID2, ARID1A* and *ARID1B*) and protein phosphatase genes (*PTPRB* and *PTPRK*) were distinctive for nail lesions in that study [28]. 

The combined use of *BRAF* and *MEK* inhibitors is one of the standards of care for patients with advanced *BRAF*-mutant melanomas. Several clinical experiences, on a limited number of patients, have shown significant responses with the use of *c-KIT* inhibitors in melanoma patients harboring mutations in exons 9, 11 or 13 [29]. A Chinese phase II study found response rates of 20–30% with prolonged progression-free survival in patients treated with imatinib [30]. Immune checkpoint blockade therapies—in particular, anti-PD-1 antibodies—showed great potential in acral melanomas: survival was longer in patients treated with immune checkpoint inhibitors as the first-line treatment than in patients receiving other systemic therapies (e.g., *BRAF/MEK* inhibitors and chemotherapy). The response rates to PD-1 blockades in patients with acral and mucosal melanomas have been comparable to the published rates in cutaneous melanoma and support the routine use of PD-1 blockades in clinical practice [31]. 

### 2.2. Scalp Melanomas 

Almost 20% of melanomas affect the head and neck region, but only a limited subset (less than 5%) arises on the scalp. Scalp melanomas appear to have a worse prognosis than melanomas in other areas of the body, probably related to the peculiar biological features of the tumor and the specific anatomy of the scalp, such as the complex lymphatic system that makes it difficult to perform a sentinel lymph node biopsy [32]. In addition, the presence of multiple layers of tissues in the scalp and the presence of the subaponeurotic space can increase the chances of tumor spread [33,34]. It has recently been reported that melanoma of the scalp with hair coverage occurs in younger individuals and has a lower Breslow thickness than melanoma of the bald scalp. Older people are usually less interested in seeking medical advice, and neoplasms tend to be more aggressive [11].

The presence itself of hair cover is an obstacle to self-examination and several cases of a diagnosis of melanoma of the scalp by hairdressers have been reported [35,36]. Dermatoscopy helps the clinician in the diagnosis, showing an atypical network, irregular streaks and a blue veil (Figure 4) [37]. In the case of the lentigo maligna subtype, rhomboidal structures, asymmetrically pigmented follicles, circle within the circle obliteration of the follicles and the finding of gray shades are observed [38,39].

The main pathogenetic factors of scalp melanomas are still unknown. The role of UV exposure is debated: chronic actinic damage is associated with the development of melanoma in patients of older age and with thinning hair, but a worse prognosis has been described in melanomas of the posterior scalp, which is usually less exposed to sunlight [11,35,40].

Scalp melanomas often present adverse histopathological features (satellite metastasis and mitoses), and the nodular variant is more often detected [40]. In the recent literature, the role of involvement of the pilosebaceous unit by neoplastic cells has been a matter of debate. According to some authors, the follicle represents a barrier to the spread of melanoma, while others consider it a poor prognostic factor [41,42,43,44,45]. It is now strongly recommended to include the presence of folliculotropism in pathological reports [46].

Specific considerations regarding the genetic background of melanomas arising on the scalp should be addressed. Somatic mutations in head and neck melanoma are not commonly observed, and the rarity of the disease makes exhaustive molecular studies difficult. Genetic data on scalp melanomas from the COSMIC database report mutations of *BRAF* in 53/141 cases (37.6%), *NF1* in 21/31 cases (67.7%), *GNA11* in 16/64 cases (25%), *GRIN2A* in 15/31 cases (48.4%) and *TP53* in 14/26 cases (53.8%) [24]. In a small series of 19 primary cutaneous head and neck melanomas, including five melanomas of the scalp, we recently found the majority of cases to be a wild type for *BRAF* and *NRAS*, with *BRAF* mutations observed in only 23% of the melanomas [47]. A retrospective study investigating 93 melanomas found a higher rate of BRAFV600E mutations (40%), while a few genetic reports of single cases showed recurrent Q209L change on the *GNA11* gene [48]. At the chromosomal level, a comparative genomic hybridization study of a nodular melanoma of the temporal scalp identified multiple chromosomal gains, including 6p, almost the entire chromosome 7 and 8q11.1-q24.3, which contain multiple important oncogenes, such as *RREB1* (6p25), *EGFR* (7p12), *BRAF* (7q34) and Myc (8q 24.21) [49]. 

The rarity of mutations in target genes represents a limitation for the treatment of stage III/IV patients with the available targeted therapies (i.e., *BRAF* and *MEK* inhibitors), reducing the treatment options. Considering the worse prognosis of scalp melanomas and the high frequency of wild-type lesions, further investigations are recommended to increase treatment options. A further understanding of new molecular signatures in melanoma progression (i.e., *PI3K* and *AKT*) is called for in the literature in order to manage nonresponsive patients [50].

## 3. Mucosal Melanomas at Uncommon Sites

### 3.1. Oral Melanomas 

Oral melanomas represent 0.5–0.7% of all oral neoplasms and 25–40% of mucosal melanomas in the head and neck region [51,52]. They account for a minority (<7%) of all melanomas, but their incidences seem to be steady (0.2 per million/year) compared to cutaneous lesions, whose incidences have been rising in the last years [51,52,53,54,55]. Oral melanomas are infrequent in Caucasian individuals (<1%), with a higher incidence among other ethnic groups, such as the Japanese (7.5%) [52,55,56,57]. They rarely appear before the fifth or sixth decades of life and seem to be more common in men than in women [51,52,53,58]. Due to their low frequency and lack of systematic studies, the etiology of oral melanomas has been poorly investigated: smoking and the resulting chronic inflammation and mechanical irritation have been addressed as a possible risk factor, but there is no definite evidence [53,58].

Oral melanomas are considered a separate entity in the melanoma scenario, different from cutaneous and ocular lesions, with their own genetic features and clinical behaviors [51]. In the majority of patients, oral melanomas develop “de novo” from non-pigmented mucosa. However, in one-third of cases, earlier oral pigmentations have been reported, including previously misdiagnosed but evolving melanoma lesions (“in situ melanomas”). A possible malignant degeneration of oral nevi has been proposed [52,54,59]. 

They are clinically characterized by brownish-to-black macules, plaque or nodules with heterogeneous shades of colors (blue, gray, red or purplish) or depigmented areas (Figure 5) [60]. The most frequent localizations are the hard palate and the alveolar mucosa [51,52,54]. Amelanotic presentations are observed in up to half of the cases, ulceration is observed in one-third of the lesions and satellite foci are not uncommon [51,52,60]. Unlike their cutaneous counterpart, the diagnostic dermoscopic criteria are not well-established, due also to the technical limitations of the complex anatomy [58,61,62]. Dermatoscopic patterns suggesting a malignant nature of the lesions include the presence of irregular diffuse pigmentation or a pseudo-network, regression structures, a blue-whitish veil and atypical vessels. Moreover, diagnostic clues are multiple colors (blue, black, gray and brown); asymmetric elements; heterogeneous borders and a sharp interruption of the reticular pattern [55,63,64]. 

For the correct management of oral pigmentations, focal lesions appearing in adults with a diameter greater than 0.5 cm and without known triggering factors or changing at follow-up should be biopsied. Nodules should be promptly excised without any monitoring [8,58]. 

However, due to the frequent lack of symptoms, the diagnosis of oral melanoma is often delayed, and the prognosis is extremely poor, with a five-year survival rate of approximately 32–35% [52,53,54,65]. In up to 43% of cases, lymph node metastases are detected at the time of diagnosis, and the risk appears to be higher for nodular lesions [53,66]. The early onset of nodal metastases is probably related to the rich vascularization and the complex lymphatic drainage of this district. Proper staging is mandatory, and magnetic resonance imaging is considered the best choice [53]. 

There are no specific guidelines or tailored treatment strategies for patients with oral melanomas. In 2009, the UICC (International Union Against Cancer, Geneva, Switzerland) 7th edition of TNM provided the classification for upper aerodigestive tract melanomas, without subgroups based on location and starting from T3 and stage III for primary melanomas arising in the epithelium/submucosa (mucosal disease) without metastases [53]. Currently, in the 8th edition of the American Joint Committee on Cancer (AJCC), there is a specific classification for mucosal melanomas of the head and neck district but none for oral lesions [67]. 

Surgery is the primary therapeutic option for oral melanomas but may be limited by the possibility of obtaining wide margins to preserve a fairly acceptable function. In the past, due to the high risk of lymph node involvement, prophylactic neck dissection was proposed [68,69]. Currently, the introduction of adjuvant therapy has profoundly changed the treatment algorithm.

The particular genetic scenario of oral melanomas has heavy implications in the therapy of metastatic diseases, and the detection of specific mutations determines the choice. Few molecular studies have focused solely on oral lesions, which are instead generally included in the wide group of mucosal melanomas. Overall, oral melanomas show a very low rate of mutations in *BRAF* (<6%) and in *GNAQ/GNA11* genes (<1%) [51,53,58]. On the contrary, *NRAS* mutations are detected in 14–22% of cases and *KIT* mutations in up to 25% [53,58]. Song et al. (2017) reported that the loss of nuclear *BAP1* expression was correlated with a poor overall survival and the development of metastases [70]. More recently, two whole-exome sequencing studies provided important insight into the mutational profile of oral melanomas [71,72]. Overall, they found a low mutation burden with an excess of the CG>AT mutation spectrum, mainly associated with the age- and smoking-related signatures. Besides the well-recognized driver mutations in the *BRAF*, *RAS* family and *KIT*, the authors also found frequent mutations in the *POLE*, *PTPRD*, *PTCHD2* and *DMXL2* genes [72]. Recurrent somatic mutations and amplifications were identified in the transmembrane nucleoporin *POM121* gene (30% of cases), which were associated with worse clinical outcomes and significantly higher tumor proliferation [71].

Regarding chromosomal aberrations, a high rate of SVs, such as inter-chromosomal and intrachromosomal translocations, between or within chromosomes 5 and 12 were detected and mainly occurred in a group of patients with worse clinical outcomes. Moreover, more than 50% of the lesions harbored recurrent focal amplifications of several oncogenes (*CDK4*, *MDM2* and *AGAP2*) at 12q13–15, and this cooccurred significantly with the amplification of TERT at 5p15 [71,72].

The low rate of *BRAF* mutations dramatically reduces the possibility of using *BRAF* and *MEK* inhibitors in oral melanoma. Therapy with imatinib is an option in patients harboring a mutation in *c-KIT*, but partial responses and the development of resistance to treatment have been described [60]. Immunotherapy can be successful, but the response rates are lower than in cutaneous melanomas. Anti-PD-1 nivolumab, combined with anti-CTLA-4 ipilimumab, appears to be more effective than nivolumab alone [73]. Clinical trials for patients with mucosal melanomas remain a key priority.

### 3.2. Genital Melanomas 

Melanomas of the genital area are extremely rare. Melanomas of the penis and scrotum represent 0.1% to 0.2% of all extraocular tumors. Penile melanomas are more frequent (69%) than scrotal melanomas (31%) and most commonly occur on the glans penis (55%), followed by the foreskin (28%), penile shaft (9%) and urethral meatus (8%) [74]. Female genitourinary tract melanomas account for 3% of all melanomas [75]. The vulva is the most frequent site involved (70%), followed by the vagina and, more rarely, the cervix [76]. Although genital melanomas can occur at any age, a mean age at diagnosis of 60 years has been reported [77,78]. The pathogenesis and risk factors have not been well-established. Exposure to UV radiation probably has no role in genital melanoma development.

The clinical presentation of genital melanomas is heterogeneous. Typically, they appear as single or multiple irregular-shaped macules, papules or nodules brownish-to-black in color (Figure 6a and Figure 7a), although, occasionally, they present as amelanotic/erythematous lesions, and ulcerations with bleeding can be present [74,79,80]. Dermatoscopy (Figure 6b and Figure 7b) is useful to identify the features such as an atypical vascular pattern, multicomponent colors with presence of a blue-whitish veil and structureless areas [77]. Melanocytic nevi, melanosis, pigmented Bowen disease and Bowenoid papulosis represent the main differential diagnoses [81]. 

Melanomas of the genital mucosa appear to be more aggressive than their cutaneous counterparts, both in male and female patients. Compared to cutaneous lesions, genitourinary melanomas have a higher median tumor Breslow thickness (3.2–12.7 mm) at diagnosis [75]. Patients with melanomas of the penis often present at an advanced stage, with the involvement of lymph nodes in the inguinal region. Aggressive tumor behaviors, a rich lymphatic vascularization of these sites and a delay in medical advice due to embarrassment or difficulty in self-observation for women have been hypothesized as the reasons for the rapid spread [75,82,83].

Surgery remains the primary treatment modality for genital melanomas; however, guidelines on surgical management are lacking. Concerns about the preservation of anatomical structures and sexual function represent a limitation to the wide surgical margins [84]. The surgery of penile melanomas varies from complete or partial amputation of the penis (full-thickness sectional excision) to organ-sparing surgery (local excision with reconstruction with a mucosal graft from the oral cavity or Mohs micrographic surgery) [77,84]. An interesting systematic review pointed out that the local recurrence rates after organ-sparing surgery and after amputation surgery were comparable, with unacceptably elevated local recurrence rates in both cases [83]. Additionally, in female genital tract melanomas, strategies of more radical surgery have been attempted in the past, although there is no evidence that a radical vulvectomy may have any benefits over local procedures [82]. Consensus guidelines need to be developed to standardize therapeutic practices in order to optimize the functional and psychosocial outcomes.

The lymph node status is the most important predictor for disease-specific survival, and patients with positive lymph nodes are now possible candidates for adjuvant treatment [80,85].

Data on the molecular alterations of genital melanomas are scarce, since the majority of genetic studies on mucosal melanomas have been performed without differentiation by anatomical site. Overall, the mutational burden of mucosal melanomas is five- to ten-fold lower than that of cutaneous melanomas, and a distinct pattern of chromosomal aberrations and a higher rate of copy number alterations have been reported [86]. Studies targeting *BRAF*, *NRAS* and *KIT* showed *KIT* as the most common mutated gene in urogenital tract melanomas, with a mutation frequency higher than 25% [87,88,89,90]. *NRAS* mutations were detected in 15% of cases, while changes in *BRAF* remained very uncommon (approximately 5%) [91,92]. A higher prevalence of *NRAS* mutations (approximately 21%) was reported in vaginal melanomas and, of *BRAF*V600E, change (33%) in vulvar cases [90,93]. In the COSMIC database, no data are available for penile and scrotum melanomas, while vulvar lesions show 10.5% of *BRAF* mutations (other than V600E). More recently, an analysis of the mutational profile by next-generation sequencing confirmed the high mutation rate of the *KIT* gene, mainly occurring in exon 11, and revealed common variants in *TP53*, *NF1* and *APC* in 22%, 19% and 10% of female genital tract melanomas, respectively. Other, less frequent mutations were found in the *CTNNB*, *GNA11*, *SDHD* and *SDHA* genes and the *TERT* promoter [94]. Finally, additional reports identified hotspot mutations in codon 625 of the *SF3B1* gene [95]. At the chromosomal level, a high rate of copy number aberrations involving gains or amplifications of chromosomes 6, 7, 8 and 11 were identified in all 13 vulvar melanomas analyzed by Yelamos et al. (2016). In these cases, the specific oncogenes targeted with the FISH probes included RREB1 at 6p25, EGFR at 7p12, BRAF at 7q34, MYC at 8q24 and CCND1 at 11q13.

There is a lack of randomized clinical trials assessing the role of systemic therapies in stage IV patients with genital melanomas. The available data mainly consist of small retrospective series and support the use of immunotherapy in improving survival. Indini et al. (2019), in a single center experience including seven female lower genital tract melanomas, reported a poor response rate for immunotherapy (28.5%) [76]. A better progression-free survival and overall survival was observed in patients under anti-PD-1 therapy compared to those receiving anti-CTLA4 (*p* = 0.01 and *p* = 0.15, respectively) [76].

The topical use of imiquimod has been reported in the literature in cases of in-situ lesions or inoperable diseases, with variable clinical responses [77,96,97]. Further studies are warranted to understand the factors improving the outcomes of these patients.

Data on adjuvant therapy for genital melanomas are relatively limited. The *c-KIT* inhibitor imatinib is the most-studied adjuvant therapy and has so far shown only a suboptimal response in trials of mucosal melanomas. Some authors reported that immunotherapy (anti-PD-1) combined with other therapies (ipilimumab, radiation therapy or imatinib) are associated with better results in the adjuvant setting than both agents alone [75,76]. 

## 4. Conclusions

Melanomas of uncommon sites deserve specific considerations regarding diagnosis and management. The prognosis is worse in most cases compared to cutaneous melanomas of other districts, and a prompt diagnosis and intervention may dramatically influence the clinical course. The rarity of mutations in genes usually found in cutaneous melanomas and currently suitable for targeted therapies requires the discovery of novel targets to develop new strategies and to change the prognosis of non-responders or wild-type patients. Future perspectives may arise from a deeper understanding of the molecular and biological mechanisms of melanomas, including microRNA expression, splicing and immunotype, in order to understand the other pathogenetic triggers and develop new target therapies [1,98,99].

## Figures and Tables

**Figure 1 jcm-10-00478-f001:**
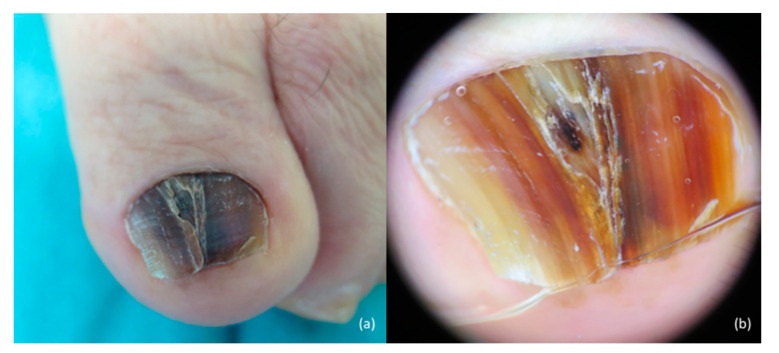
Nail melanoma of the great toenail clinically presenting as a longitudinal melanonychia (**a**). At onychoscopy, (**b**) heterogeneous irregular longitudinal bands of pigment with nail plate dystrophy; pigmentation extends to the hyponychium (Hutchinson’s sign).

**Figure 2 jcm-10-00478-f002:**
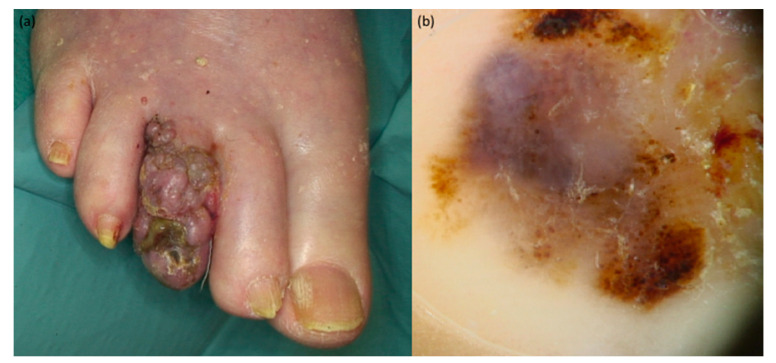
Nail melanoma of the 3rd digit of the foot: (**a**) clinical presentation showing neoplastic cutaneous spread to the periungual tissues and Hutchinson’s sign: an exophytic and verrucous growth is observed. Dermoscopy (**b**) reveals bluish, black and milky-red structureless areas and irregular brown dots.

**Figure 3 jcm-10-00478-f003:**
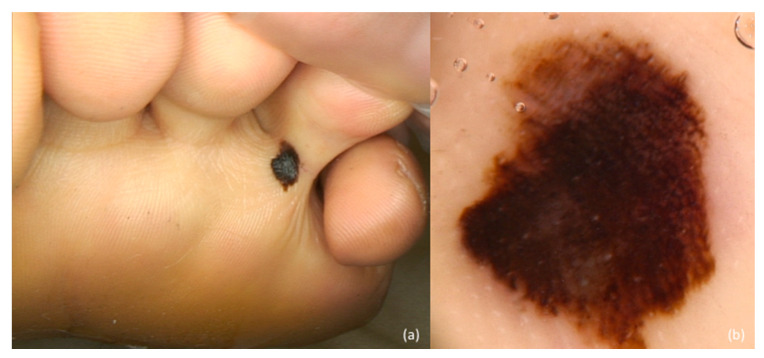
Plantar melanoma. Clinical presentation (**a**) reveals a black irregular macule. Dermoscopy (**b**) shows a parallel ridge pattern and structureless dark brown and black areas.

**Figure 4 jcm-10-00478-f004:**
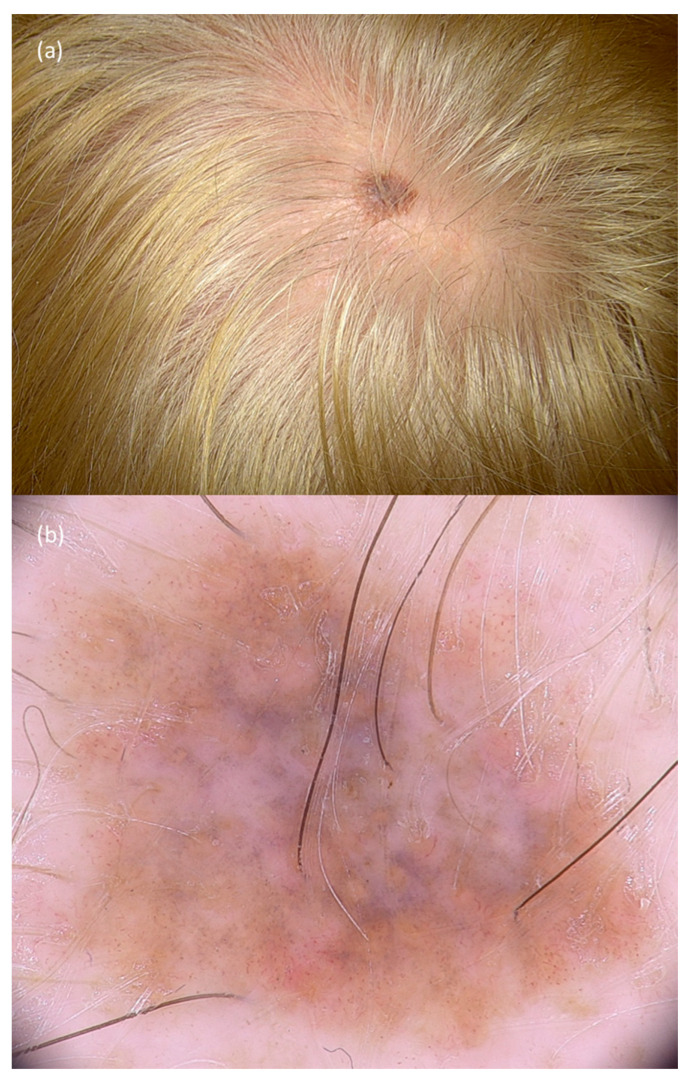
Scalp melanoma: clinical presentation (**a**) showing a brownish-to-bluish macule. Dermoscopy (**b**) revealed an irregular brown pseudo-network with a central blue veil and regression areas.

**Figure 5 jcm-10-00478-f005:**
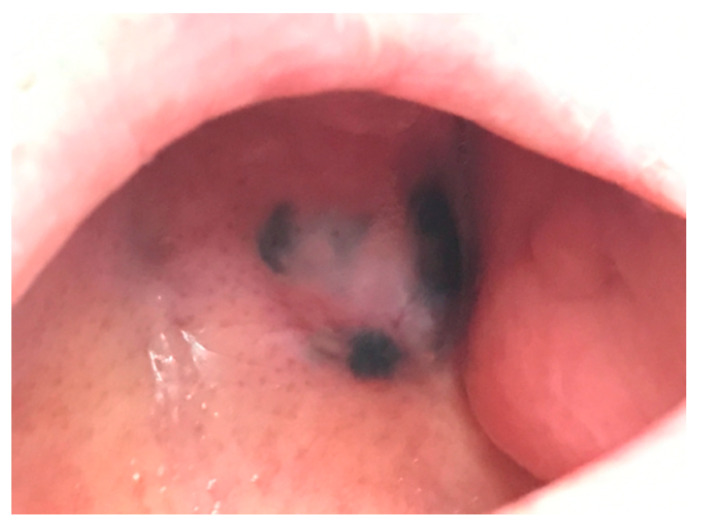
Oral melanoma: clinical presentation showing a black nodule with depigmented areas.

**Figure 6 jcm-10-00478-f006:**
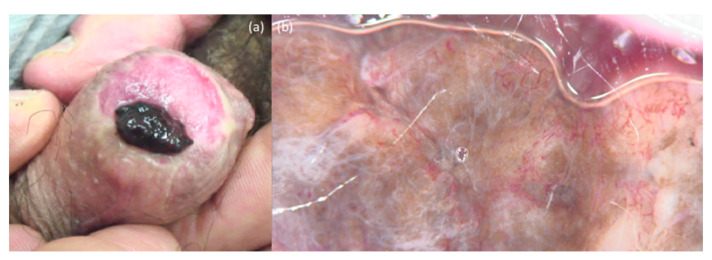
Penis melanoma. The clinical presentation (**a**) shows a black nodule of the glans. Dermoscopy (**b**) reveals polymorphous vessels with structureless brownish and white areas.

**Figure 7 jcm-10-00478-f007:**
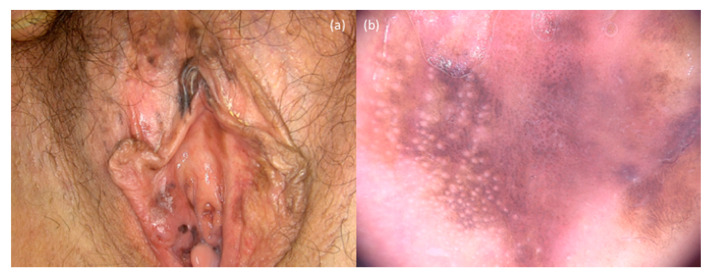
Vulvar melanoma. The clinical image (**a**) shows multiple irregular-shaped brown/black macules. Dermoscopy (**b**) reveals black/blue and white structureless areas, regression areas, irregular blue/gray dots and an atypical brownish digitiform pattern.

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
