# Peer review of "Cutaneous and Mucosal Melanomas of Uncommon Sites: Where Do We Stand Now?"

_jcm, 2021, doi:10.3390/jcm10030478_

Round 1

Reviewer 1 Report

In the manuscript jcm-1081970 entitled CUTANEOUS AND MUCOSAL MELANOMAS OF UNCOMMON SITES: WHERE DO WE STAND NOW? the authors review clinical as well as genetic characteristics of melanomas arising in unusual sites, namely selected mucosal and cutaneous locations. Established and emerging new therapies are also discussed. The manuscript is well organized, transparent and intelligible, but the overall linguistic aspect of the text should be improved (see specific comments below). The conclusion of the article is clearly stated, that is – melanomas of uncommon sites are entities with distinctive clinical and molecular features and require scientific attention in order to develop desperately needed new treatment options. The manuscript is well referenced with predominantly recent literature, including the authors’ own work. In this reviewer’s opinion, several flaws need to be addressed before publication of the article:

L20 The article ‘an’ is mistakenly used before ‘localizations’ (plural noun).

L21 In the first sentence the authors aim to generally define ‘melanomas arising at uncommon sites’. The use of ‘and/or’ is misleading, because difficult histopathological interpretation per se is not sufficient to name a lesion ‘melanoma arising at uncommon site’. The location, in this context, should be the defining feature. Histopathological interpretation of such lesions is often challenging (as may be in the case of a clinically ‘ordinary’ melanoma), but diagnostic difficulties are not essential in the definition. This conceptual error is replicated in the introduction (LL 34-37). Please correct these sections.

L26 The sentence ‘The available treatment…’ is put in between two sentences directly referring to melanomas of uncommon sites and thus seems out of context. Please adjust this section.

LL39-41 The sentence ‘The term “sun-ptorected…’ does not explain the nature of criticism of the term ‘sun-protected melanomas’ clearly enough. The authors of the referenced article (Shi K. et al) state that it is inadequate to use an umbrella term ‘sun-protected melanomas’ to describe a heterogenous pool of mucosal, acral and vulvovaginal melanomas. Please rephrase.

L108 Please change ‘variant’ to plural ‘variants’.

L132 Repetition ‘report’, ‘reporting’. Please rephrase.

L175 The use of a comparative form ‘greater’ suggest a juxtaposition of two types of melanoma. If so, what is scalp melanoma compared to?

L180 The adjective ‘poor’ is not suitable to describe histopathological features. Please rephrase.

L188 Please move ‘difficult’ to the end of the sentence.

L192 Please change ‘melanoma’ to plural ‘melanomas’.

L201 ‘The rarity of mutations in target mutations represent’ is clearly a mistake. Please rephrase.

LL 250-256 The authors mention 2009 UICC TNM and state that no specific staging for oral mucosal lesions is present in the 8th ed. AJCC. This is misleading as one might think that oral melanoma is completely ignored by AJCC staging. Rather, the authors should describe how such lesions are staged according to the current guidelines (chapter “Mucosal Melanoma of the Head and Neck” in the 8th ed. of AJCC manual)

L273 Please use a comma instead of a period after ‘KIT’.

L300 It is not clear to which subject the word ‘their’ refers to. Please rephrase.

L362 Please use ‘consist of’ instead of ‘consist in’.

L383 Please replace ‘target’ with ‘targeted’.

LL396-408 These sections are unnecessary extracts from the journal guidelines.

The quality of figures/photographs is poor (Figure 2 seems acceptable). Please provide figures of the best possible quality.

The authors should consider a professional linguistic support as, in this reviewer’s opinion, it would substantially improve the quality of the manuscript.

Author Response

Dear Reviewer,

We are submitting to your attention the revision of the manuscript entitled CUTANEOUS AND MUCOSAL MELANOMAS OF UN-COMMON SITES: WHERE DO WE STAND NOW?”.

The modifications have been performed according to the requirements and have been highlighted in yellow in the text.

We herein report the responses to the comments:

L20 The article ‘an’ is mistakenly used before ‘localizations’ (plural noun).

Dear Reviewer, we apologize for the mistake and we corrected it.

L21 In the first sentence the authors aim to generally define ‘melanomas arising at uncommon sites’. The use of ‘and/or’ is misleading, because difficult histopathological interpretation per se is not sufficient to name a lesion ‘melanoma arising at uncommon site’. The location, in this context, should be the defining feature. Histopathological interpretation of such lesions is often challenging (as may be in the case of a clinically ‘ordinary’ melanoma), but diagnostic difficulties are not essential in the definition. This conceptual error is replicated in the introduction (LL 34-37). Please correct these sections.

Dear Reviewer, thank you for your suggestion. We rephrased the sentence as follows: “Melanomas of uncommon sites encompass both cutaneous and mucosal lesions related to an unusual localization in specific ethnic groups. The histopathological interpretation may be challenging and known site-related atypical features may sometimes increase diagnostic dilemmas for pathologists”.

L26 The sentence ‘The available treatment…’ is put in between two sentences directly referring to melanomas of uncommon sites and thus seems out of context. Please adjust this section.

Thank you, we rephrased the sentence.

LL39-41 The sentence ‘The term “sun-ptorected…’ does not explain the nature of criticism of the term ‘sun-protected melanomas’ clearly enough. The authors of the referenced article (Shi K. et al) state that it is inadequate to use an umbrella term ‘sun-protected melanomas’ to describe a heterogenous pool of mucosal, acral and vulvovaginal melanomas. Please rephrase.

Dear Reviewer, we agree with you. We rephrased the sentence as follows: “The umbrella term “sun-protected melanomas” has been considered inadequate to de-scribe a heterogenous pool of mucosal, acral and vulvovaginal melanomas with distinct genetic profiles [1]. Treatment choice and outcome may therefore be challenging”.

L108 Please change ‘variant’ to plural ‘variants’.

Thank you, we changed as required.

L132 Repetition ‘report’, ‘reporting’. Please rephrase.

Thank you, we changed as required

L175 The use of a comparative form ‘greater’ suggest a juxtaposition of two types of melanoma. If so, what is scalp melanoma compared to?

Dear Reviewer, we decided to delete the sentence in order to avoid a misunderstanding.

L180 The adjective ‘poor’ is not suitable to describe histopathological features. Please rephrase.

Thank you, we changed with “adverse”

L188 Please move ‘difficult’ to the end of the sentence.

Thank you, we changed as required

L192 Please change ‘melanoma’ to plural ‘melanomas’.

Thank you, we changed as required

L201 ‘The rarity of mutations in target mutations represent’ is clearly a mistake. Please rephrase.

Thank you, we rephrased the sentence as follows: “The rarity of mutations in target genes represents a limitation for the treatment of stage III/IV patients with the available targeted therapies..”

LL 250-256 The authors mention 2009 UICC TNM and state that no specific staging for oral mucosal lesions is present in the 8th ed. AJCC. This is misleading as one might think that oral melanoma is completely ignored by AJCC staging. Rather, the authors should describe how such lesions are staged according to the current guidelines (chapter “Mucosal Melanoma of the Head and Neck” in the 8th ed. of AJCC manual)

Dear Reviewer, we agree with you and we changed as follows: “Currently, in the 8th edition of the American Joint Committee on Cancer (AJCC), there is a proper classification for mucosal melanomas of the head and neck district, but not a specific focus on oral lesions“.

L273 Please use a comma instead of a period after ‘KIT’.

Thank you, we changed as required

L300 It is not clear to which subject the word ‘their’ refers to. Please rephrase.

Thank you, we changed with “genital melanoma”

L362 Please use ‘consist of’ instead of ‘consist in’.

Thank you, we changed as required

L383 Please replace ‘target’ with ‘targeted’.

Thank you, we changed as required

LL396-408 These sections are unnecessary extracts from the journal guidelines.

Thank you, we changed as required

The quality of figures/photographs is poor (Figure 2 seems acceptable). Please provide figures of the best possible quality.

We provided figures/photographs with a higher quality. However, we believe that a reduction of quality may be due to the fact that photos have to be pasted in the word file. We are available to send them as separate files.

The authors should consider a professional linguistic support as, in this reviewer’s opinion, it would substantially improve the quality of the manuscript.

Dear Reviewer, the paper has been revised by a linguistic specialist.

Best regards

The Authors

Reviewer 2 Report

This is a well written manuscript reporting updated findings of melanomas affecting “uncommon” sites.

There are a few points to be checked as mentioned below.

1) Although these melanomas are “uncommon” in white persons, they are “common” in non-white persons. The authors correctly described this in the 2nd paragraph of the section “2.1 Aral melanoma” in page 2.

2) Clear clinical and dermoscopic images are illustrated. It may be better to describe briefly the important findings in these images.

3) page 1, Introduction, 2nd paragraph:

The authors describe that these melanomas are “aggressive” variants. This may be incorrect. For example, there are several papers reporting that the prognosis of acral melanoma is not worse than other subtypes of melanoma, if stratified by the tumor thickness.

4) page 6, “3.1 Oral melanoma”, 2nd paragraph:

The authors describe “a possible malignant degeneration of oral nevi has been proposed.” However, there is a possibility that the preceding lesions are not nevi but evolving melanoma lesions (“in situ melanomas”). It may be better to mention this possibility.

5) page 8, “3.2 Genital melanoma”, 2nd line of the 1st paragraph:

Is the description of “0.1-0.2% of all extraocular tumors” correct? Is it not “extraocular” but “genital”?

Author Response

Dear Reviewer,

We are submitting to your attention the revision of the manuscript entitled CUTANEOUS AND MUCOSAL MELANOMAS OF UN-COMMON SITES: WHERE DO WE STAND NOW?”.

The modifications have been performed according to the requirements and have been highlighted in yellow in the text.

We herein report the responses to the comments:

This is a well written manuscript reporting updated findings of melanomas affecting “uncommon” sites.

There are a few points to be checked as mentioned below.

1) Although these melanomas are “uncommon” in white persons, they are “common” in non-white persons. The authors correctly described this in the 2nd paragraph of the section “2.1 Aral melanoma” in page 2.

Dear Reviewer, thank you for your suggestion. We rephrased the sentence in order to clarify it.

2) Clear clinical and dermoscopic images are illustrated. It may be better to describe briefly the important findings in these images.

Dear Reviewer, thank you for your suggestion. We provided a brief description.

3) page 1, Introduction, 2nd paragraph:

The authors describe that these melanomas are “aggressive” variants. This may be incorrect. For example, there are several papers reporting that the prognosis of acral melanoma is not worse than other subtypes of melanoma, if stratified by the tumor thickness.

Dear Reviewer, thank you for your suggestion. We rephrased the sentence in order to soften it.

4) page 6, “3.1 Oral melanoma”, 2nd paragraph:

The authors describe “a possible malignant degeneration of oral nevi has been proposed.” However, there is a possibility that the preceding lesions are not nevi but 3). It may be better to mention this possibility.

Dear Reviewer, thank you for your suggestion. We mentioned this possibility.

5) page 8, “3.2 Genital melanoma”, 2nd line of the 1st paragraph:

Is the description of “0.1-0.2% of all extraocular tumors” correct? Is it not “extraocular” but “genital”?

Dear Reviewer, the sentence “0.1-0.2% of all extraocular tumors” is correct, as a number of authors reported this kind of description in their papers.

Best regards

The authors